# Immunometabolic Therapeutic Targets of Graft-versus-Host Disease (GvHD)

**DOI:** 10.3390/metabo11110736

**Published:** 2021-10-27

**Authors:** Kudakwashe Mhandire, Komalpreet Saggu, Nataliya Prokopenko Buxbaum

**Affiliations:** Roswell Park Comprehensive Cancer Center, Department of Pediatrics, Buffalo, NY 14203, USA; kudakwashe.mhandire@roswellpark.org (K.M.); komalpreet.saggu@roswellpark.org (K.S.)

**Keywords:** GvHD, T cells, metabolism, therapeutic targets, glycolysis, fatty acid oxidation

## Abstract

Allogeneic hematopoietic stem cell transplantation (allo-HSCT) is a curative option in the treatment of aggressive malignant and non-malignant blood disorders. However, the benefits of allo-HSCT can be compromised by graft-versus-host disease (GvHD), a prevalent and morbid complication of allo-HSCT. GvHD occurs when donor immune cells mount an alloreactive response against host antigens due to histocompatibility differences between the donor and host, which may result in extensive tissue injury. The reprogramming of cellular metabolism is a feature of GvHD that is associated with the differentiation of donor CD4+ cells into the pathogenic Th1 and Th17 subsets along with the dysfunction of the immune-suppressive protective T regulatory cells (Tregs). The activation of glycolysis and glutaminolysis with concomitant changes in fatty acid oxidation metabolism fuel the anabolic activities of the proliferative alloreactive microenvironment characteristic of GvHD. Thus, metabolic therapies such as glycolytic enzyme inhibitors and fatty acid metabolism modulators are a promising therapeutic strategy for GvHD. We comprehensively review the role of cellular metabolism in GvHD pathogenesis, identify candidate therapeutic targets, and describe potential strategies for augmenting immunometabolism to ameliorate GvHD.

## 1. Introduction

Allogeneic hematopoietic stem cell transplantation (allo-HSCT) has the potential to cure multiple hematological disorders, including aggressive malignancies [1]. The curative potential of allo-HSCT, the graft-versus-leukemia (GvL) effect, can be mediated by several immune subsets [2,3,4]. However, these cell populations may also trigger graft-versus-host disease (GvHD), a morbid and prevalent barrier to allo-HSCT [5,6]. CD4+ T cells are at the center of GvHD pathogenesis. Activated CD4+ T cells differentiate into a plethora of cytokine-secreting T helper (Th) cells including Th1, Th2, Th9, Th17, and regulatory T cells (Tregs) [7,8,9]. Pro-inflammatory Th1 and Th17 CD4+ cells direct GvHD targeting the skin, lungs, gastrointestinal tract, and other tissues [4,10,11,12]. By contrast, regulatory CD4+ Forkhead box protein P3 (Foxp3) + cells (Tregs) are capable of ameliorating GvHD through immunosuppressive mechanisms [13,14,15,16,17]. Alloreactivity induces a metabolic shift in immune cells from a quiescent oxidative phosphorylation (OXPHOS) dominant state to an anabolic profile featuring aerobic glycolysis as the main energy source. In addition, glutamine and fatty acid metabolism provide intermediates for the tri-carboxylic acid (TCA) cycle which fuels the OXPHOS activity seen in early T cell activation [10,18,19,20]. The increased glycolytic flux supports the proliferation of the pathogenic Th1 and Th17 cells. The resultant high T effector/Treg ratio facilitates GvHD pathogenesis in target tissues [9,10,13,21].

There are several forms of GvHD, including acute, chronic, late acute, and overlap syndrome, and some of the biological and pathophysiological features are shared by these entities [4,5]. Acute GvHD exhibits systemic inflammation and tissue destruction in the skin, liver, and gastrointestinal tract that involves alloreactive donor T-cell-mediated cytotoxicity [4,11]. The pathophysiology of cGvHD is complex and thought to involve the interplay between dysregulated innate and adaptive immune cell populations, as well as tissue fibrosis and organ damage [6,22]. Acute GvHD is the main risk factor of chronic GvHD and together they negatively impact on the success of allo-HSCT [23,24,25,26].

Standard GvHD prophylactic measures involve peri-transplant immunosuppression directed primarily at the T cells compartment, including anti-thymocyte globulin (ATG), alemtuzumab, alpha-beta depletion, post-transplant cyclophosphamide (PTCy), calcineurin inhibitors, and methotrexate [27,28,29,30,31]. Despite prophylaxis, the prevalence of acute and chronic GvHD is high. First-line therapeutic interventions aim to provide global immune suppression via the administration of systemic corticosteroids with or without calcineurin inhibitors [27,32]. Unfortunately, these measures increase the risk of infection, relapse, and secondary malignancies [32,33]. In addition, systemic corticosteroids have many side effects and are ineffective for many patients with acute and chronic GvHD, with clinical responses to second-line agents being also being limited [34,35]. An ideal GvHD therapeutic intervention would suppress pathogenic allogeneic immune reactivity while preserving the graft-versus-leukemia (GvL) effect and maintain functional anti-infection immunity [2,36]. Given that malignant cells share cellular metabolism features with alloreactive T cells, metabolic reprogramming represents a promising strategy for the therapeutic targeting of GvHD [7,8,37]. Thus, this review focuses on the immune metabolism of GvHD and the potential metabolic pharmacological targets of GvHD.

## 2. T Cell Metabolism in Allo-HSCT

Naïve T cells primarily depend on oxidative phosphorylation. Following allo-HSCT, donor T cells are stimulated by mismatched recipient cell antigens to undergo pro-glycolytic metabolic reprogramming and form allogeneic T effector cells (Teffs) [38,39,40]. During this phase, pyruvate is directed towards lactate synthesis under the influence of pyruvate dehydrogenase kinase 1 (PDHK1) [18,41]. The upregulation of glycolysis supports cell proliferation as glycolytic intermediates feed anabolic pathways, including the pentose phosphate pathway (PPP) [9,42,43]. T cell receptor (TCR) antigen-presenting cell (APC) interaction stimulates the phosphoinositide 3-kinases (PI3K)-protein kinase B-mechanistic target of rapamycin (mTOR) pathway to stabilize the glycolytic microenvironment. Furthermore, protein kinase B (a.k.a. Akt) activates glycolytic enzymes such as hexokinase (HK) and phosphofructokinase (PFK) through phosphorylation [44,45]. Akt also upregulates the expression of the transmembrane glucose transporters (Glut)1 and Glut3 in alloreactive T cells [46,47].

Concomitantly, pyruvate is decarboxylated into acetyl-CoA which is relayed into the TCA cycle to facilitate NADH and flavin adenine dinucleotide (FADH_2_) synthesis. In the presence of oxygen, NADH and FADH2 fuel ATP generation by shuttling electrons along the electron transport chain in OXPHOS [43,48]. Studies show that both glycolysis and OXPHOS are upregulated in alloreactive cells [20,49]. However, the increase in OXPHOS activity is transient and may characterize early events in GvHD pathogenesis and T cell activation [49,50], perhaps preceding the development of tissue hypoxia. The alloreactive metabolism and cell proliferation form a hypoxic environment that is characterized by the elevated expression of hypoxia-induced factor 1 alpha (HIF-1α) [51,52]. HIF-1α is a transcription factor, which modulates the expression of glycolytic enzymes, such as HK1, and glucose transporters, namely Glut1, and Glut3 [19,51]. The messenger ribonucleic acid (mRNA) expression of other glycolytic enzymes and transporters, such as glyceraldehyde 3-phosphate dehydrogenase *(Gapdh)*, lactate dehydrogenase (*Ldh)-a*, monocarboxylate transporter *(Mtc4)* and phosphoglycerate kinase *(Pgk1)* have also been found to be elevated in alloreactive T cells in GvHD [19,53].

The mTOR is a family of nutrient sensors that direct Th1 and Th17, Treg, and memory T (Tem) cell differentiation [54]. The main forms of mTOR are mTORC1 and mTORC2. mTORC1 is the prominent pro-glycolysis member, which directs cell proliferation in the effector phase of GvHD. mTORC1 induces the expression of interferon (IFN)-γ, tumor necrosis factor (TNF), and perforins, which drive tissue damage in GvHD [45,55,56]. In addition to glycolysis, alloreactive T cells also use glutamine and fatty acids as energy sources. In GvHD, glutamine metabolism is an alternative carbon source for the tricarboxylic acid (TCA) cycle [57,58,59,60]. The uptake of glutamine in alloreactive T cells is enabled by the upregulation of glutamine transporters alanine serine cysteine transporter (ASCT)1, ASCT2, and l-leucine transporter (LAT)1 [61]. Glutamine generates α-ketoglutarate (α-KG), which enters the TCA cycle where citrate is generated and channeled towards lipid synthesis. Glutamine metabolism is also a source of the reducing agents NADH and NAPH for ATP production [57,58,59]. 

The rising adenosine diphosphate (ADP)/ adenosine triphosphate (ATP) ratio in low cellular energy conditions activates the nutrient sensor adenosine monophosphate-activated kinase (AMPK). AMPK inhibits the acetyl CoA carboxylases (ACC)1 and ACC2, expressed in the cytoplasm and mitochondria, respectively [62]. ACC1 and ACC2 convert acetyl-CoA to malonyl-CoA. The inhibition of ACC2 results in a decrease in malonyl-CoA, which signals carnitine palmitoyltransferase 1a (CPT1a) and facilitates the shuttle of long chain fatty acids (LCFA) into the mitochondria for fatty acid oxidation (FAO) [57,62,63,64]. FAO produces acetyl-CoA and generates reduced nicotinamide adenine dinucleotide (NADH) and FADH_2_ for eventual ATP production [65]. ACC1 in the cytosol produces malonyl-CoA as a precursor for de novo fatty acid synthesis (FAS). Thus, ACC2 and ACC1 regulate FAO and FAS, respectively [65,66]. Pathogenic Th17 cells rely on de novo FAS mediated by ACC1 for differentiation [67]. Tregs, on the other end, depend more on FAO than glycolysis for energy generation [68]; hence, promoting FAO may ameliorate GvHD. 

## 3. T Cell Metabolic Targets for GvHD 

### 3.1. Glycolysis Targets

The metabolic reprogramming from OXPHOS towards glycolysis sustains the proliferation and secretion of pro-inflammatory cytokines in alloreactive T cells [6,8,43]. Our group confirmed through transcriptomic, protein, and metabolic analyses that the CD4 T effector memory (Tem) cells, a pathogenic subset that mediates GvHD in target tissues, is highly glycolytic [53], providing a further rationale for testing glycolytic inhibitors in GvHD. Pharmacological inhibitors of glycolytic enzymes have been previously tested in an aGvHD pre-clinical model [19] and identified glycolysis as an important metabolic target for re-programming alloreactive T cells in aGvHD (Figure 1). 

A potent glycolytic inhibitor, 2-deoxy-d-glucose (2-DG) acts by competitively mimicking D-glucose, resulting in the inhibition of HK1, HK2, and glucose-6-phosphate isomerase [69,70]. Given 2-DG’s ability to efficiently inhibit glycolysis, it has been incorporated into combinatorial treatment approaches for cancer with cisplatin [71], and metformin [72]. 2-DG has been shown to suppress IFN-γ–secreting CD4+ and CD8+ T cell proliferation and ameliorate GvHD in major histocompatibility (MHC) mismatched pre-clinical models [19]. However, additional studies are needed to clearly delineate 2-DG’s effects in GvHD, as Nguyen et al. found low efficacy with short-term and improved efficacy with long-term use of 2-DG, and the latter was unfortunately associated with increased toxicity [19]. 

Another promising glycolytic target for the control of GvHD is 6-phosphofructo-2-kinase/fructose-2, 6-bisphosphatases (PFKFB3). PFKFB3 converts fructose 6-phosphate to fructose 1,6-biphosphate and is the rate-limiting factor in glycolysis. PFKFB3 is elevated in tumors, where it facilitates cell proliferation through increased glycolytic flux [73,74]. Accordingly, PFKFB3 inhibitors, such as 3-(3-Pyridinyl)-1-(4-pyridinyl-2-propen-1-one (3PO) and 1-(4-pyridinyl)-3-(2-quinolinyl)-2-propen-1-one (PFK15) have been shown to exhibit anti-cancer activity [75,76]. 3PO attenuates T cell proliferation by interfering with glucose influx, lactate secretion, and TNF-α secretion [77]. The inhibition of glycolysis by 3-PO and PFK15 intervention notably alleviated GvHD in an MHC-mismatched (B6-> BALB/c) allo-HSCT mouse model [19]. In these studies, allogeneic recipients receiving anti-PFKFB3 therapies demonstrated improved survival and lower GvHD scores. 

The blockade of the nutrient sensor mTOR through the use of its antagonist, rapamycin (sirolimus), is another key aspect of metabolic reprogramming in GvHD. mTORC1 is sensitive to rapamycin inhibition, while mTORC2 is not [54]. Following TCR stimulation, mTORC1 activity is upregulated by the increased uptake of amino acids [78] and increased glucose in the intracellular space [79]. mTORC1 activation supports overall aerobic glycolysis and mTOR-deficient T cells fail to differentiate into Th1, Th2, and Th17 lineages [56]. Furthermore, mTOR stabilizes glycolytic flux favoring HIF-1α and Myc [80,81]. Thus, mTOR antagonists may play a role in the treatment of GvHD. Rapamycin also interferes with the expression of the glucose transporters, Glut1 and Glut3, and amino acid transporters [38]. Accordingly, the clinical use of rapamycin for GvHD prophylaxis and treatment has been reported [82,83,84,85]. Pre-clinical efficacy was demonstrated when rapamycin was administered shortly after MHC-mismatched (B6 -> BALB/c) allo-HSCT and continued until day 14, resulting in lower GvHD scores and reduced levels of proinflammatory cytokines [19]. In the same study, mRNA expression of glycolytic enzymes was significantly reduced following rapamycin administration. Rapamycin treatment has also been shown to result in CD4 Treg expansion and downregulation of pathogenic Th1 and Th17 cells in an MHC-mismatched (C57BL/6-> BALB/c) mouse model [86,87]. A multicenter randomized clinical trial of patients with lymphoma undergoing allo-HSCT following reduced-intensity conditioning (RIC) reported a lower incidence of acute GvHD in the sirolimus/rapamycin arm than in the tacrolimus and methotrexate arm [88]. However, rapamycin monotherapy exhibits poor efficacy and is associated with adverse events, including elevated serum creatinine, hemolytic uremic syndrome, and nephrotic syndrome [85,89]. Rapamycin given in combination with other drugs may provide a more effective GvHD treatment [90,91]. Rapamycin analogs are also propitious candidates for GvHD therapy. The administration of GS-649443 (PI3K specific inhibitor) from day 28 post-allo-BMT was at effective at treating sclerodermatous cGvHD in a minor histocompatibility mismatch (B10.D2-> BALB/c) mouse model. GS-649443 administration was also associated with improved skin and overall GvHD scores and was associated with downregulation of the Th17 compartment [92]. 

HIF-1α upregulates the expression of glucose transporters (Glut1 and Glut3), glycolytic enzymes (Hk, Gaphd), and IFN-γ [51,93]. In the MHC-mismatched (C57BL/6-> BALB/c) model, the inhibition of HIF-1α using echinomycin prolonged post-allo-HSCT survival and reduced GvHD scores [52]. In the same study, echinomycin treatment was associated with increased Treg numbers in lymphoid organs. Importantly, echinomycin treatment preserved GvL function leading to prolonged leukemia-free survival. Thus, the pharmacological inhibition of HIF-1α represents a promising treatment strategy for GvHD. However, further insight into the role of HIF-1α in GvHD is warranted since another animal study demonstrated a contrasting effect of HIF-1α, i.e., its stabilization via the prolyl hydroxylase inhibitor, dimethyl oxalyl glycine (DMOG), led to improved survival in the same MHC mismatch model (C57BL/6-> BALB/c) [94]. Notably, the administration of DMOG attenuated T cell infiltration of the gut and downregulated proinflammatory cytokine expression. However, the authors did not report on the Th17/Treg dynamics; hence, the favorable GvHD outcomes could have been associated with HIF-1α’s general anti-inflammatory function. Thus, additional data on the use of HIF-1α inhibition in the GvHD are needed to assess its role on T cell proliferation vs. tissue injury dynamics. Hypothetically, HIF-1α inhibition could work well in prevention of aGvHD through dampening early glycolytic activity but could have limited efficacy once hypoxia is established in the highly inflamed target tissues when stabilization of HIF-1α may be beneficial for dampening inflammation induced injury.

While the role of glycolysis in promoting alloreactive cell activity and tissue injury is well described, alloreactive T-cells also depend on OXPHOS to meet their ATP demand for proliferation [49]. In our ex vivo Seahorse analysis of pathogenic CD4+ T effector cells, we found a concomitant high extracellular acidification rate (ECAR) and oxygen consumption rate (OCR), suggestive of high rates of both, glycolysis and OXPHOS, respectively [53]. More extensive metabolic profiling by Gatza et al. also found that alloreactive The T cells show an increase in mitochondrial activity especially in the early period post-allo-HSCT. In a haploidentical (C57BL/6-> B6D2F1) model, they showed that alloreactive T cells, but not other proliferating cells or bone marrow cells, upregulated both OXPHOS and glycolysis pointing to the important role of OXPHOS in alloreactive cell survival [20]. Such T cells also showed hyperpolarized mitochondrial membrane potential and elevated superoxide production. The pharmacological inhibition of mitochondrial activity using the mitochondrial F_0_F_1_ ATPase inhibitor Bz-423 in two mouse models, minor HC mismatch [C3H.SW to B6] and major HC mismatch [BALB/c to B6], resulted in the attenuation of GvHD via the induction of alloreactive T cell apoptosis [20]. Thus, mitochondrial OXPHOS in donor alloreactive cells may be an important therapeutic target for GvHD. 

### 3.2. Fatty Acid Metabolism Targets in GvHD

Th17 cells rely on de novo FAS to produce cell membrane phospholipids, while the immunosuppressive Treg cells depend on FAO for energy [1,2]. Due to its catalytic role in malonyl CoA production for de novo FAS, ACC1 is a potential target for GvHD intervention (Figure 1). The inhibition of ACC1 starves Th17 cells of the FAS precursor. In the MHC-mismatched (C57BL/6-> BALB/c) model, T cells deficient in ACC1 were remarkably less pathogenic than the wild-type T cells [57]. Tregs were also significantly higher in the recipients of ACC1-deficient T cells, indicating the active role of ACC1 in GvHD. The blunted GvHD phenotype was maintained when recipient mice received the pharmacological ACC1 inhibitor, soraphen A (SorA), demonstrating the potential therapeutic the role of ACC1 inhibition in GvHD. Furthermore, both ACC1 genetic deficiency and SorA administration dampened T cell glycolytic activity. Additionally, another pre-clinical study demonstrated the ability of ACC1 inhibition in attenuating GvHD [67]. ACC1 inhibition has also been successfully explored in ischemic cerebral infarction, a condition in which CD4+ T cell mediated inflammation propagates tissue injury. SorA administration significantly reduced ischemic brain injury and preserved the balance of peripheral Treg/Th17 cells by dampening glycolysis and FAS in peripheral CD4+ T cells in a mouse model of ischemic stroke [66].

Empirical evidence shows that the oxidation of LCFA supports Treg survival [21,95]; hence, promoting FAO has the potential to ameliorate GvHD. Metformin stabilizes AMPK leading to the enhancement of FAO, which supports Treg differentiation. Accordingly, metformin is an effective agent for the treatment of GvHD [62]. In an MHC mismatched model (C57BL/6-> BALB/c), metformin attenuated the clinical manifestations of aGvHD and improved survival [96]. Metformin treatment reduced Th1 and Th17 cells and promoted Tregs. In combination with the immunosuppressant tacrolimus, metformin administration resulted in reduced GvHD severity [97]. Another group reported a reduction in Th1 and Th17 cells and an increase in Treg cells in five patients who underwent liver transplantation and were on a daily regimen of metformin (1000 mg/day dose) for diabetes mellitus. Another AMPK agonist, AICAR (5-aminoimidazole-4-carboxamide ribonucleotide) was shown to enhance Treg cell expansion through cellular mitochondrogenesis and increased fatty acid uptake and oxidation [98]. 

Metformin, however, activates AMPK phosphorylation indirectly by repressing ATP synthesis through inhibiting Complex I of the electron transport chain [99]. Metformin is therefore capable of controlling GvHD by limiting energy supply to proliferating cells regardless of AMPK activity. The inhibition of the ETC by metformin is also detrimental to the survival of other immune cells and has been shown to slow pathogenic cell proliferation in other inflammatory conditions [97,100,101,102]. AMPK activity without metformin intervention modulates T cell proliferation and alloreactivity. A recent study demonstrated that AMPK is dispensable in early TCR stimulation and acute proliferation but is required for sustained Teff proliferation [103,104]. This is supported by the finding that AMPK genetic knockout donor T cells reduced GvHD severity in an MHC mismatch (C57BL/6 to BALB/c) model [103], demonstrating the independent role of AMPK in GvHD development. Furthermore, the transplantation of T cells deficient in both AMPKα1 and AMPKα2 lessened GvHD without attenuating the anti-leukemia effect [105]. Further research is warranted to explore other ways, beyond metformin administration, through which AMPK could be modulated in the treatment of GvHD. Targeting FAO (Figure 1) for therapeutic intervention is a balancing act. For instance, the pharmacological inhibition of CPT1a using etomoxir is expected to suppress FAO, which would interfere with Treg cell differentiation [106] and could be expected to exacerbate GvHD. However, etomoxir was explored as a potential therapy in GvHD and was reported to suppress alloreactive T cells in a haploidentical model (B6-> B6D2F1), demonstrating anti-GvHD activity [95]. These findings could not be replicated in another mouse study, where the inhibition of FAO by etomoxir did not show an effect on donor T cell proliferation [19]. A subsequent study reported that Treg differentiation may not rely on CPT1a activity and/or AMPK activated FAO [106]. These findings suggest that the use of etomoxir in GvHD warrants further study. Moreover, without substantive evidence probing the genetic blockade of FAO to reveal its independent influence on Treg differentiation, the potential role of FAO in GvHD development should be further delineated.

### 3.3. Glutamate Metabolism Targets

In alloreactive T cell proliferation, the TCA cycle supplies intermediates for anabolic pathways that sustain cell proliferation [9]. To maintain the TCA cycle, its intermediates are replenished by a biochemical flux of precursors rerouted from anaplerotic pathways such as glutaminolysis [59]. Glutamine is the main amino acid in plasma. It is converted to *α*-ketoglutarate to replenish citric acid intermediates NADH, citrate, and malate. These glutamine-derived carbon products can be converted to oxaloacetate, a precursor to the pentose phosphate pathway (PPP) [107]. An in vitro analysis of T cells isolated after allo-HSCT in mice showed an upsurge in the TCA cycle and glutamine-dependent anaplerosis [108]. Glutamine metabolism is also required for Th cell differentiation [58]; for instance, glutamine transporter SLC7A-deficient T cells fail to differentiate upon activation [8]. Furthermore, the expression of genes encoding glutamine-regulating enzymes was found to be significantly higher in allogeneic CD4+ and CD8+ T cells than in syngeneic T cells [19]. Given these data, glutamine metabolism is a potential therapeutic strategy (Figure 1) for curbing alloimmune T cell proliferation. 

Currently, there is insufficient evidence linking glutamine metabolism inhibition with GvHD outcomes. However, inhibiting glutamine metabolism with 6-diazo-5-oxo-L-norleucine (DON) in combination with rapamycin suppressed Th17 proliferation and attenuated arthritis in a mouse model of the latter [109]. Cancer cells also rely on glutamine metabolism to fuel their proliferation. As such, the suppression of glutaminase, an enzyme that initiates the breakdown of glutamine, was shown to have anti-tumoral activity by starving the cancer cells of glutamine metabolites [110,111]. mTOR promotes cell utilization of glutamine during proliferation and mTOR blockade via rapamycin ameliorates GvHD’s clinical severity [83,84,85,88,112]. 

Interestingly, glutamine supplementation has also been implemented for GvHD therapy. Intraperitoneal glutamine administration in a mouse study inhibited GvHD-induced inflammation and tissue injury, improved clinical scores and survival [113]. While intracellular glutamine metabolism may support pathogenic T cell proliferation in GvHD, target tissue cells in the gastrointestinal tract also rely on glutamine for survival under stress from allogeneic immune attack [113]. This was demonstrated in a haploidentical (C57BL/6-> B6D2F) model in which oral glutamine supplementation was associated with reduced jejunum TNF-α expression and reduced histological GvHD scores in the same tissue [114]. With these equivocal findings, glutamine targeting in GvHD warrants further investigation. 

## 4. Antigen-Presenting Cell Metabolism in GvHD

While T cells are key cellular mediators of GvHD pathogenesis, APCs play an active role in the development of GvHD [115,116]. Dendritic cells (DCs), macrophages, and B cells process alloantigens and present them on major histocompatibility complex (MHC) molecules to the TCR [117]. The MHC–TCR interaction initiates a cascade of signals that results in T cell activation. In response to stimuli, APCs secrete cytokines that regulate the immune response through pro- and/or anti-inflammatory signals [116,117,118]. The following section discusses DC and macrophage metabolism and its potential targeting to attenuate GvHD. 

### 4.1. Dendritic Cell Targets 

Dendritic cells are highly efficient APCs equipped with specialized endocytosis to capture, process, and present antigens for recognition by T cells [119,120]. The maturation of naïve DCs is initiated by the detection of pathogen-associated molecular patterns (PAMPs), in case of infection, or damage-associated molecular patterns (DAMPs) in inflammatory conditions such as GvHD [121]. DC maturation involves changes in metabolic and transcription programming to sustain the bioenergetic demands of immune activation. In the mature state, DCs induce T-cell differentiation into pro-inflammatory (Th1, Th2, Tfh, and Th17), and regulatory (Tregs) subsets [122,123]. This cell differentiation plays a key role in GvHD pathogenesis. DAMPs generated during chemoradiotherapy conditioning stimulate host DCs to provoke donor T-cell-mediated GvHD in the liver, colon, and skin [116,118,124]. Furthermore, MHC-mismatch-driven CD4+ T-cell-dependent GvHD can be induced by both donor and host DCs. Mature activated DCs, in an inflammatory allo-HSCT microenvironment, have increased bioenergetic needs in order to sustain anabolic metabolism and facilitate migration into the target tissues [125]. These bioenergetic demands are met by reprogramming from mitochondrial β-oxidation to lipid metabolism and OXPHOS to glycolysis [126,127]. 

The glycolytic inhibitor 2-DG has been shown to attenuate DC maturation and to induce T cell activation [127]. The administration of 2-DG in a mouse study decreased the expression of DC co-stimulatory molecules CD80 and CD54, which is indicative of the downregulation of DC activation [127]. In the same study, 2-DG also suppressed DC’s ability to activate CD4+ and CD8+ T-cells. Human plasmacytoid DCs stimulated by a viral infection in the presence of 2-DG exhibit suppressed co-stimulatory activity [63]. DCs upregulate HIF-1α to sustain glycolysis in inflammatory conditions [128]. HIF-1α inhibition has been reported to ameliorate GvHD through blunting T cell differentiation [52], while it may also have exerted anti-inflammatory effects through DC reprogramming. Similarly, 2-DG for ameliorating GvHD through T cell metabolic reprogramming has been explored [19], but it may also be postulated to exhibit some of its effects in GvHD through suppressing DC function. DCs use stored intracellular glycogen for energy generation during proliferation-induced maturation [129]. The glycogen phosphorylase inhibitor, CP91149, attenuates TLR-mediated DC maturation and impairs DC’s ability to initiate lymphocyte activation. Therefore, it may offer therapeutic potential in the setting of GvHD [129]. Glutaminolysis, under the influence of mTOR, also plays a role in DC maturation, as the inhibition of glutaminolysis and OXPHOS have been demonstrated to interfere with pDC activation [63]. Thus, HIF-1α and glutaminolysis inhibition form potential targets for DC modulation-based GvHD therapy; however, relevant studies have not yet been reported. 

While DC maturation can directly influence GvHD through the activation of donor T cells, the inhibition of DCs should be carefully studied, especially in the setting of allo-HSCT for malignant indications. Elze et al. reported moderate-to-severe acute GvHD 60 days post-allo-HSCT in 45 children with leukemia who exhibited low pDC reconstitution, as measured in circulation. By contrast, high pDC reconstitution was associated with a higher risk of relapse [130]. These findings caution against strategies to suppress pDCs without further probing the question of whether GvL and GvHD’s effects can be separated, and whether target tissue resident vs. circulating DCs perform distinct functions in mediating these effects. In conclusion, subsets of DCs may play distinct roles in GvHD biology and are an active area of study [131].

### 4.2. Macrophage Targets 

Macrophages (MCs) are phagocytic antigen-presenting cells with dynamic profiles that reciprocally differentiate into pro-inflammatory M1 and anti-inflammatory M2 subsets [132]. The M1 macrophages are induced by the detection of foreign antigens and pro-inflammatory cytokines (TNF-α, INF-γ), leading to cytokine secretion and the activation of Th1 cells, and the release of reactive nitrogen and oxygen species [133]. By contrast, M2 cells are induced by inflammatory cytokines (IL-4 and IL-13) and Th2 cells to mostly induce immune regulation and the resolution of inflammation [134]. M1 and M2 MCs feature distinct metabolic profiles [132]. The M1 profile is dependent on glycolysis for energy generation, and enforces two breaks in the TCA cycle to produce citrate and succinate [135]. The accumulated citrate generates itaconate, an important antimicrobial agent [136]. Succinate stabilizes HIF-1α, which sustains the glycolytic metabolism of M1 cells [81,137]. Glycolysis feeds the PPP to generate NADPH, which acts as a co-factor of iNOS in NO production [138]. Acetyl-CoA generated by glycolysis is used in FAS. Enhanced FAS is an important feature of M1 metabolism. HIF-1α also suppresses OXPHOS in M1 macrophages, while M2 macrophages rely on OXPHOS for ATP generation. Additionally, in contrast to M1, M2 MCs rely on the TCA cycle [139]. In M2 MCs, FAO and glutamine metabolism are upregulated to provide intermediates for the TCA cycle. Glycolysis and, subsequently, the PPP, are downregulated in M2 macrophages [133]. 

Macrophage infiltration of target tissues has been associated with GvHD. The level of infiltration may be directly proportional to the severity of GvHD [140,141]. The complex role of macrophages in GvHD is exemplified by observations of high M1/M2 ratios in aGvHD, and the reverse (low M1/M2 ratio) in cGvHD [115]. Thus, M1 and M2 macrophages may be pathogenic in aGvHD and cGvHD, respectively [142]. The secretion of nitric oxide (NO) through NO synthase (iNOS) by M1 cells in inflamed tissue is a plausible mechanism for promoting aGvHD [143]. Furthermore, macrophages facilitate alloreactive T cell responses to host antigens [144]. Active M1 macrophages tend to favor Th17 cell differentiation while suppressing Treg cells, thus delineating their potential role in GvHD pathogenesis. Injecting donor-derived M2 macrophages has been shown to attenuate GvHD and prolong survival following murine allo-HSCT [145]. In cGvHD, the interaction of M2 macrophages with CD4+ cells and fibroblasts may fuel tissue fibrosis [146]. 

Metabolic reprogramming aimed at dampening M1 polarization and overall macrophage reduction could alleviate aGvHD. The metabolic inhibitors, 2-DG and 3-PO, as well as rapamycin, which have been reported to attenuate GvHD through T cell metabolic reprogramming and modulation of CD4+ T cell differentiation [19,56,69,86,87,92], are also candidates for macrophage downregulation in GvHD therapy given M1’s dependence on glycolysis and M2’s dependence on OXPHOS, respectively. While 2-DG is expected to inhibit M1 polarization, 2-DG lacks specificity and has been reported to interfere with M2 polarization too [70], perhaps in a glycolysis-independent manner. HIF-1α inhibitors may also play a role in modulating MC metabolism for dampening GvHD [52], and based on what is known about M1 MC metabolism, they could be hypothesized to inhibit this subset. An in vitro study showed that rapamycin dampens macrophage activation [147] and another reported that rapamycin induced apoptosis in M2 macrophages but not in M1 [148], which is consistent with the known metabolic utilization patterns of M2 MCs described above. 

The plasticity of macrophage subsets in inflammation and disease presents a challenge to the development of metabolic targets for augmenting macrophage function in GvHD. Macrophages in inflammatory conditions exhibit M1 polarization, while those in tumors are typically M2 tumor-associated macrophage (TAM)-like responses [132]. Thus, macrophage-targeting cancer therapeutics are directed at dampening M2 TAM while promoting pro-inflammatory M1 macrophages [149]. While hypothetically, this could be beneficial in ameliorating cGvHD, a concern would be the potential negative impact on GvL as well as the potential of this approach to upregulate the pro-glycolytic and inflammatory M1 cells associated with aGvHD. These studies provide cautionary evidence for targeting macrophage metabolism in malignant allo-HSCT settings. 

### 4.3. Myeloid-Derived Suppressor Cell Targets 

The emerging role of myeloid-derived suppressor cells (MDSC) in dampening alloreactivity provides a rationale for exploring this cell subset to potentially ameliorate GvHD. While their classification as myeloid cells and/or APCs is debatable, MDSCs are characterized by their immune-suppressive activity in tumor and inflammatory conditions [150,151]. Thus, enhancing MDSCs’ suppressive function could ameliorate inflammation associated with GvHD. For example, infusion of ex-vivo-generated MDSCs was shown to offer transient efficacy against alloreactivity [152]. ATP released from tissue during conditioning binds to the P2x7R on MDSCs resulting in inflammatory activation, which extinguishes MDSC immune suppression and ignites a proinflammatory cascade that fuels aGvHD [153]. The genetic (P2x7 knockout) and pharmacological (A-438079) inhibition of P2x7 was demonstrated to promote recipient survival in an MHC mismatched allo-HSCT study [154]. MDSC metabolism could be explored as a strategy for GvHD, but could be potentially limited in malignant allo-HSCT settings due to the concerns already described in the MC-targeting section.

### 4.4. B Cell Ttargets

B cells detect foreign antigens and elicit immune responses through antibody production, complement activation, and antibody-mediated cellular toxicity among other mechanisms [155,156]. B cells are also capable of mounting antibody-independent immune regulatory activities involving cytokine secretion and antigen presentation via MHC. Consequently, B cells play a substantive role in chronic GvHD pathogenesis and other autoimmune disorders [157,158,159]. B lymphopoiesis occurs in the bone marrow, where a substantial amount of autoimmune naïve B cells arises by random immunoglobulin gene rearrangement. B cells bearing autoreactive BCRs are directed towards apoptosis via negative selection [156,160]. B cell activating factor of the TNF family (BAFF) directs B cell development, differentiation, and function [157,161]. Elevated BAFF expression promotes B cell metabolism and has been shown to increase allo- and autoreactive B cell populations [162,163]. During immune reconstitution following allo-HSCT, excess levels of BAFF disrupt the negative selection of alloreactive B cells and increase their BCR hyperresponsiveness [164,165]. The presence of auto- and allo-antibodies in patients with cGVHD is indicative of a loss of B cell tolerance [166,167], which may fuel GVHD pathogenesis [168]. B cells from cGVHD patients have been shown to exhibit a heightened metabolic state and resistance to apoptosis [157]. Activated B cells, similar to T cells, favor a glycolytic profile [169], and glucose uptake is upregulated upon B cell receptor (BCR) activation [170,171]. BAFF stimulates the PI3K, Akt and mTORC1 pathways to enhance glycolysis [172,173], while glutamine metabolism is also upregulated in activated B cells [174,175].

B-cell-directed targeting in GvHD has been performed via B cell depletion, BAFF inhibition, and BCR signal inhibition. The benefits of rituximab therapy for GvHD have been demonstrated in several clinical studies [176,177,178]. B cell proliferation after BCR activation is achieved through Bruton’s tyrosine kinase (BTK) and spleen tyrosine kinase (SYK) stimulation; hence, several BTK and SYK inhibitors are undergoing clinical testing for GvHD [30,179,180], given encouraging pre-clinical data. Importantly, ibrutinib was the first FDA-approved second-line treatment for cGvHD after several studies demonstrated clinical efficacy [180,181,182,183]. Immunometabolic targets involving glycolysis and glutamine have not yet been reported as potential treatment strategies for augmenting B cell function in GvHD. However, the possibility of simultaneous B cell inhibition by pharmacological agents that have been successfully used to target T cell metabolism is present and warrants investigation.

## 5. Potential Caveats in Metabolic Targeting of Immune Cells in GvHD

The potential of metabolic targets in the treatment of GvHD may be somewhat hindered by a lack of specificity. Pharmacological agents aimed at dampening the differentiation and proliferation of pathogenic Th1 and Th17 cells may interfere with Treg differentiation and function [184,185]. As an example, mTOR inhibition using rapamycin is widely reported to suppress glycolytic activity in allogeneic T cells. However, mTOR is also required for the differentiation of protective Tregs [184,185]. Rapamycin administration also disrupts fatty acid and cholesterol synthesis, which are essential for Treg cell development [186]. The AMPK agonist, metformin, enhances Treg cell differentiation by promoting FAO. However, it can also inhibit the electron transport chain (ETC) to the detriment of other cells, including GvL-facilitating Tem cells; thus, this metabolic intervention may negatively impact GvL. Similarly, the Cpt1a inhibitor, etomoxir, ameliorates GvHD in murine models but also interferes with the ETC [187]. Targets such as HIF-1α may play competing roles in GvHD, with HIF-1α potentially promoting GvHD by enhancing glycolysis (a key pathway for alloreactive immune cells), but it may also prevent tissue damage in the hypoxic microenvironment associated with GvHD. Thus, both the inhibition and the promotion of HIF-1α have been reported to alleviate GvHD in pre-clinical studies [52,94,188]. Despite these challenges, the prospects for metabolic targeting in the clinical management of GvHD are encouraging. The demonstration of the efficacy of multiple pharmacological agents in preclinical studies opens the door for clinical trials. The preservation of GvL that is critical to the success of allo-HSCT in cancer is dispensable in non-cancerous conditions for which allo-HSCT is curative, including thalassemia, sickle cell anemia and other hemoglobinopathies [189,190,191], making the implementation of metabolic targets in GvHD therapy more achievable. The pharmacological modulation of metabolism has also been successfully explored in the treatment of autoimmune conditions [102], which further supports testing in the setting of allo-immunity.

## 6. Conclusions

Immune cell metabolism plays a major role in the pathogenesis of GvHD. T cell differentiation is an important aspect of GvHD pathogenesis with glycolysis fueling alloreactive proliferation. Glycolysis sustains the anabolic activities of both T cells and antigen-presenting cells. A key feature of pro-GvHD metabolism is the upregulation of Th1, Th2 and Th17 cell differentiation and the suppression of Tregs. The increase in glutaminolysis and FAS also supports the pathogenic cell subsets, while FAO and OXPHOS increase protective Treg cells. Pre-clinical studies using metabolic inhibitors of glycolysis and fatty metabolism have generated evidence supporting these therapeutic options. Glycolytic inhibitors, FAS inhibitors, and FAO promoters ameliorate GvHD by restraining pathogenic cell proliferation and enhancing immune-regulating subsets. Those agents that reduce GvHD severity in animal studies typically demonstrate suppression of the pathogenic Th17 cells, along with enhancement of Tregs. The potential of immune metabolic reprogramming for the treatment of GvHD is propitious but has not yet been explored in the clinical setting, while it is beginning to be developed for other immune-mediated conditions [192,193].

## Figures and Tables

**Figure 1 metabolites-11-00736-f001:**
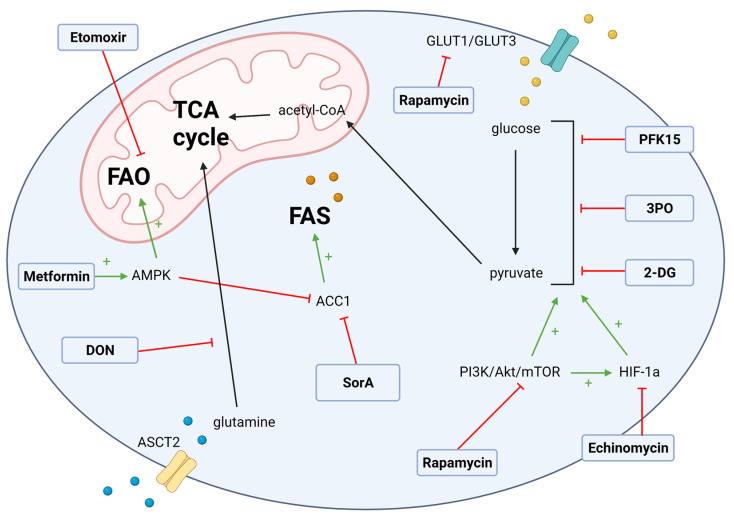
Potential T cell metabolism therapeutic targets in GvHD. Alloreactive T cells undergo a shift in metabolism characterized by an upregulation of glycolysis, along with increased reliance on glutamine and fatty acid metabolism. Regulators of these metabolic pathways serve as promising drug targets for the treatment of GvHD. The upregulation of glycolysis is mediated by the transcription factor HIF-1α and the nutrient sensor mTOR, which may be targeted by echinomycin and rapamycin, respectively. The drugs PFK15, 3PO, and 2-DG also interfere with glycolysis. PFK15 and 3PO are both inhibitors of PFKFB3, and 2-DG inhibits the glycolytic enzymes hexokinase and glucose-6-phosphate isomerase. Fatty acid metabolism is regulated by the acetyl CoA carboxylases and the nutrient sensor AMPK. Soraphen A is an ACC1inhibitor, which promotes FAO while suppressing fatty acid synthesis. Metformin is an AMPK agonist, which similarly upregulates fatty acid oxidation. Etomoxir is an inhibitor of fatty acid oxidation and may offer therapeutic benefits in GvHD. Glutamine metabolism may be another promising pathway for therapeutic targeting in GvHD. DON, an inhibitor of glutamine-utilizing enzymes, has been shown to suppress Th17 proliferation in mice, but its therapeutic benefits have yet to be tested in the context of GvHD. ACC1, acetyl coenzyme A carboxylase 1; acetyl-CoA, acetyl coenzyme A; Akt, Protein Kinase B; AMPK, adenosine monophosphate-activated protein kinase; ASCT2, alanine/serine/cysteine transporter 2; 2-DG, 2-deoxy-d-glucose; DON, 6-diazo-5-oxo-l-norleucine; FAO, fatty acid oxidation; FAS, fatty acid synthesis; Glut1/Glut3, glucose transporter 1/glucose transporter 3; HIF-1α, hypoxia inducible factor 1-alpha; mTOR, mechanistic target of rapamycin; PFK15, 1-(4-pyridinyl)-3-(2-quinolinyl)-2-propen-1-one; PI3K, phosphoinositide 3-kinase; 3PO, 3-(3-pyridinyl)-1-(4-pyridinyl-2-propen-1-one); SorA, Soraphen A; TCA cycle, tricarboxylic acid cycle.

## Data Availability

No new data were created or analyzed in this study. Data sharing is not applicable to this review.

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
