# Peer review of "Immunometabolic Therapeutic Targets of Graft-versus-Host Disease (GvHD)"

_metabolites, 2021, doi:10.3390/metabo11110736_

Round 1
Reviewer 1 Report
In this review authors give a detailed description of especially of T cells, but also of APCs, metabolic reprogramming after allogenic hematopoietic stem cell transplant, highlighting how these metabolic changes are involved in Graft Versus Host Disease (GvHD). The aim of the authors is to give broad knowledge of the therapeutic strategies that target metabolic pathway, the aim of which is to avoid the onset of GvHD.
The review is quite nice and well written. Several typographic errors throughout the manuscript should be corrected
Addressing the following issues would significantly enhance its predicted appeal:
In the first part of “introduction” section (lines 24-38), authors should make a more discursive and less schematic description, integrating the sentences together. Furthermore, the concept of metabolic shifts in alloreactivity should be introduced more clearly.
In the line 66-74 they should create a more discursive connection between phrases and specify that these are alloreactive T cells. In the line 83 there is no need to rewrite mTOR name in full.
The end of part 3.2 is a bit incomplete.
In the 4.2 part (lines 349-367) they should re-organize the speech by first describing the macrophages and the difference between M1 and M2 and then the metabolic changes associated with the graft.
In the 4.4 part (lines 420-431) it is necessary to remodel the speech, describing first B cells and then BAFF with its function and any therapeutic target
In the line 17 they use FAO, without specify extensive name. In the line 39 they should write aGvHD and cGvHD next to acute and chronic, respectively. At line 215 they can use only FAS, without extensive name. In the part 4 title, it would be better to remove APC and use the abbreviation in the text (at line 292) In the line 368 specify abbreviation MCs
Typos need to be corrected (extra spaces at line 160, 267, 280).
My final assessment of this manuscript will depend on the revision of the specific points above reported.
Reviewer 2 Report
In their review, authors seek to highlight metabolic targets during the initiation and maintenance of graft-versus-host disease (GVHD). The review is overall well written and the structure is appropriate for the topic and covered in sufficient detail. The challenge with the review in its current format is that there exist multiple areas in the study of post-transplant metabolism where data from different groups land on two sides of an issue. In discussing a couple of these instances in their review, authors do not cite all of the relevant studies, leading to the potential for a biased interpretation of the actual biology. In some cases, the authors do a nice job of presenting a balanced story, for example when describing the therapeutic potential of 2-deoxyglucose (first full paragraph on page 4), summarizing comments on Hypoxia inducible factor-1alpha and GHVD (first full paragraph on page 5), or in outlining the potential suppression of plasmacytoid dendritic cells (last paragraph on page 7). However, as highlighted below, other situations would benefit if the data were presented more broadly so that readers could make their own interpretation.
- In the first paragraph of the introduction, authors write “Alloreactivity induces a metabolic shift in immune cells from oxidative phosphorylation (OXPHOS) to aerobic glycolysis”. However, there is ample evidence from early studies that OXPHOS increases in alloreactive T cells early post-transplant (Gatza et al. Sci Trans Med (2011) 3:67ra8) and that treatment with an inhibitor of the electron transport chain to decrease OXPHOS actually ameliorates GVHD. Furthermore, the larger paradigm of immune cell metabolism envisions OXPHOS versus glycolysis as an “either/or” scenario (either the cell adopts OXPHOS or they do glycolysis). However, there is evidence that alloreactive T cells can increase both glycolysis and oxygen consumption simultaneously (Brown and Byersdorfer, Front Immunol (2020) 11:1517). Thus, just because T cells increase glycolysis doesn’t mean they can’t also increase OXPHOS at the same time. Finally, recent papers have suggested that the glycolysis paradigm may be less relevant in vivo as T cells activated in vivo have been shown to rely more heavily on OXPHOS than their in vitro stimulated counterparts (Ma et al. Immunity 51:856). These points, as well as their references, should be incorporated into the review.
- In the bottom paragraph of page 5 (under Fatty acid metabolism targets in GVHD, staring around line 233), authors highlight a number of studies involving metformin and conclude that metformin must be functioning through upregulation of AMPK signaling. However, metformin indirectly increases AMPK phosphorylation by inhibiting the electron transport chain. Thus, activation of AMPK could be a correlative event and not a causative one. Authors note this point about metformin later in the review, but it deserves inclusion here as well (or perhaps only here). This emphasis is particularly important in light of two recent publications that demonstrate that a lack AMPK in donor T cells decreases the severity of GVHD without impacting graft-versus-leukemia effects (Lepez et al. Sci Rep 2020 10:21673, Monlish et al. JCI Insight. 2021 6:e143811). These papers should also be highlighted in the paragraph discussing metformin and AMPK activation and their implications discussed accordingly.
- In at least two places in their review, authors highlight or suggest that “promoting FAO (fatty acid oxidation) has the potential to ameliorate GVHD”. However, FAO increases in donor T cells in some GVHD models and definitive, genetic blockade of FAO has never been shown. Thus, the role of blocking versus promoting FAO remains controversial. As such, statements should be broadened to include both sides of this argument without preference given to one side over the other without offering definitive proof. In a similar vein, there are multiple references in this review which state that regulatory T cells require increased FAO. However, much of this early data has been called into question recently because mice lacking carnitine palmitoyl transferase 1a (CPT1a), which are capable of only minimal FAO, continue to generate Treg at the same rate as WT cells (Raud B et al. Cell Metab (2018) 28:504). Therefore, the role of FAO during GVHD is best described as an evolving area.
Minor:
There are a few instances of miscalled references.
- The callout out for Reference 63 (line 108, page 3) describes “upregulation of ACC1 and other FAS enzymes” “associated with poor GHVD outcomes”. However, the paper cited actually highlights the role of FAO in GVHD and did not imply a role for FAS.
- On line 109-110, authors note “FAO upregulation has been shown to ameliorate GVHD” then cite reference 65. However, reference 65 is the Raud et al paper noted above, which describes results using T cells from CPT1a KO mice but has no data using GVHD model. These lines should either be re-written or the correct reference appropriated.
- On line 264-265, authors write “In vitro analysis of T cells isolated after AHSCT in mice showed an upsurge in the TCA cycle and glutamine-dependent anaplerosis” and then cite reference 100. However, reference 100 is a human study. I believe the authors intended to cite reference 99 again, but this should be reviewed and corrected as necessary.
Authors, whenever possible, should also take care to note whether a particular study used a model of chronic versus acute GVHD, as the pathophysiology of these two entities is likely distinct and could have implications on interpretation of the findings.
Reviewer 3 Report
Peer review on Mhandire et al: “Immunometabolic therapeutic targets of graft-versus-host disease (GvHD)” for metabolites.
This is an extensive review on the metabolic basis of inflammation and the resulting possible targets for the treatment of graft versus host disease (GVHD). The aspects of possible opposing effects pharmacologic-metabolic interventions in different immune cell subsets, as well the issue of co-targeting the (often necessary) graft-versus-leukemia effect, are critically discussed. The review is very well written.
I have only a few, predominantly minor, comments:
General:
The term “AHSCT” should be replaced by “allogeneic HSCT”, or “alloHSCT” or “alloSCT”, since AHSCT is frequently used to abbreviate autologous HSCT.
Page 4 (lines 183-185):
The following report should be cited in the context of renal complications of rapamycin treatment:
Hochegger K, Wurz E, Nachbaur D, Rosenkranz AR, Clausen J. Rapamycin-induced proteinuria following allogeneic hematopoietic stem cell transplantation. Bone Marrow Transplant. 2009 Jul;44(1):63-5. doi: 10.1038/bmt.2008.433. Epub 2009 Jan 19. PMID: 19151795.
Page 5 (line 234):
The reference #58 (Garcia and Shaw, Molecular Cell, 2017) should support metformin being an “effective agent for the treatment of GvHD”, however, this was not addressed in the cited paper. It might better be stated that the paper suggests metformin being a promising agent for the management of GVHD.
